# Sequential Release of Panax Notoginseng Saponins and Osteopractic Total Flavone from Poly (_L_-Lactic Acid) Scaffold for Treating Glucocorticoid-Associated Osteonecrosis of Femoral Head

**DOI:** 10.3390/jfb14010031

**Published:** 2023-01-04

**Authors:** Guiyu Feng, Pingxin Zhang, Jian Huang, Yao Yu, Fenghe Yang, Xueqian Zhao, Wei Wang, Dongyang Li, Song Sun, Xufeng Niu, Limin Chai, Jinyu Li

**Affiliations:** 1Key Laboratory of Chinese Internal Medicine of Ministry of Education and Beijing, Dongzhimen Hospital, Beijing University of Chinese Medicine, Beijing 100700, China; 2Department of Orthopedic, Dongzhimen Hospital, Beijing University of Chinese Medicine, Beijing 100700, China; 3Key Laboratory of Biomechanics and Mechanobiology (Beihang University), Ministry of Education, Beijing Advanced Innovation Center for Biomedical Engineering, School of Biological Science and Medical Engineering, Beihang University, Beijing 100083, China; 4Yuquan Hospital Affiliated to Tsinghua University, Beijing 100040, China

**Keywords:** Poly (_L_-lactic acid), osteopractic total flavone, panax notoginseng saponins, osteogenesis, angiogenesis

## Abstract

Glucocorticoids inhibit angiogenesis in the femoral head, which fails to nourish the bone tissue and leads to osteonecrosis. Restoring angiogenesis is not only essential for vessel formation, but also crucial for osteogenesis. Poly (_L_-lactic acid) (PLLA) is commonly used in the bone tissue engineering field. Panax notoginseng saponins (PNS) and osteopractic total flavone (OTF) promote angiogenesis and osteogenesis, respectively. We designed a sequentially releasing PLLA scaffold including PLLA loaded with OTF (inner layer) and PLLA loaded with PNS (outer layer). We assessed the osteogenic effect of angiogenesis in this scaffold by comparing it with the one-layered scaffold (PLLA embedded with OTF and PNS) in vivo. Results from the micro-CT showed that the data of bone mineral density (BMD), bone volume (BV), and percent bone volume (BV/TV) in the PO-PP group were significantly higher than those in the POP group (*p* < 0.01). Histological analyses show that the PO-PP scaffold exhibits better angiogenic and osteogenic effects compared with the one-layered scaffold. These might result from the different structures between them, where the sequential release of a bi-layer scaffold achieves the osteogenic effect of vascularization by initially releasing PNS in the outer layer. We further explored the possible mechanism by an immunohistochemistry analysis and an immunofluorescence assay. The results showed that the protein expressions of vascular endothelial growth factor (VEGF) and platelet endothelial cell adhesion molecule-1(CD31) in the PO-PP scaffold were significantly higher than those in the POP scaffold (*p* < 0.01); the protein expressions of osteocalcin (OCN), osteopontin (OPN), and alkaline phosphatase (ALP) in the PO-PP scaffold were significantly higher than those in the POP scaffold (*p* < 0.05). Upregulating the expressions of angiogenic and osteogenic proteins might be the possible mechanism.

## 1. Introduction

Glucocorticoid-associated osteonecrosis of the femoral head (GONFH) is known as bone death, resulting from the use of chronic glucocorticoids [1]. Glucocorticoids could directly damage endothelial cells, impair new blood vessels formation, and decrease the vascular supply to bone tissue [2]. The vascular supply to the femoral head involves lateral and medial circumflex femoral arteries, which penetrates the femoral head and supply most of the femoral epiphysis. Vascular damage easily leads to the inability to deliver nutrients to the bone tissue of the femoral head, leading to femoral head necrosis [1]. Restoring angiogenesis is not only essential for vessel formation, but also crucial for osteogenesis [3,4]. Angiogenesis and osteogenesis, closely related to each other, are both crucial for bone repair.

Bone tissue engineering is a promising strategy for bone repair [5,6]. Biomaterials represent a key element in bone tissue engineering. Poly (_L_-lactic acid) (PLLA), a highly versatile biodegradable aliphatic polyester, is commonly used as degradable biomaterial [7]. It has been widely used in biomedical and surgical fields, such as surgical equipment and fusion substance, tissue engineering scaffolds, and drug carriers including orthopedic treatment, owing to its safety and low toxicity [8,9,10]. The most commonly used PLLA in tissue engineering has been approved by FDA [11]. Recent medical studies confirmed that the implant of an electrospun, silicon-doped vaterite/PLLA membrane used to treat the calvaria defect in the rabbit model could enhance the proliferation of pre-osteoblasts and promote new bone formation [12]. Implanted PLLA also could improve the attachment, proliferation, and differentiation of pre-osteoblasts, and lead to the effective osteo-conductivity contributing to new bone formation in vitro and in vivo [13,14].

In addition to biomaterials, drug delivery systems comprise another key element in bone tissue engineering. The sequential treatment of collagen scaffolds with bone morphogenetic protein-2 (BMP-2) following stromal cell derived factor-1 (SDF-1) showed superior new bone regeneration compared with their simultaneous treatment [15]. The initial release of BMP-2, followed by the sequential release of alendronate (ALN) in collagen-hydroxyapatite composite scaffold, significantly increases osteogenic activity owing to the synergistic effect of BMP-2 and ALN. In this study, we aim to obtain a sequentially releasing PLLA scaffold, which enhances the osteogenic effect by restoring angiogenesis.

Panax notoginseng saponins (PNS) is the main ingredient of Panax Notoginseng (a medicinal herb). Treatment by PNS promotes angiogenesis, as PNS activates the VEGF-KDR and PI3K-Akt-eNOS pathways, which contribute to angiogenesis [16]. Furthermore, PNS induces the production of VEGF-A, VEGFR-1, and VEGFR-2 in bone marrow mesenchymal stem cells (BMSCs) [17]. Osteopractic total flavone (OTF) is the main ingredient of Drynariae Rhizoma (a medicinal herb). Several studies confirmed that OTF could improve the formation of new bone. OTF in vivo experiments could significantly promote the formation of epiphyses and bone trabeculae, increase bone mineralization, accelerate the differentiation and mineralization of osteoblasts, and inhibit the bone resorption of osteoclasts [18,19,20,21]. In this study, we constructed a bi-layered PLLA scaffold including PLLA embedded with OTF (PO, inner layer) and PLLA embedded with PNS (PP, outer layer). We hypothesize that PNS in the outer layer would be released initially, followed by the OTF release of the inner layer. We further presumed that this sequential release model could enhance the osteogenic effect by promoting angiogenesis. 

We designed this sequentially released PO-PP scaffold to adopt the advantage of this sequential release model: PNS being initially released, followed by the release of OTF. We also assessed the osteogenic effect of angiogenesis (a key advantage of sequential release) in the PO-PP scaffold by comparing it with a one-layered PLLA scaffold (PLLA embedded with OTF and PNS) in vivo. We hypothesized that this sequentially released scaffold (PO-PP) could enhance the osteogenic effect by restoring angiogenesis at the basis of initially released PNS-promoting angiogenesis.

## 2. Material and Methods

### 2.1. Preparation of One- and Bi-Layer Scaffolds

Poly (_L_-lactic acid) (PLLA, molecular weight: 2.00 × 10^5^) was purchased from Evonik Industries AG (Essen, German). 1,4-Dioxane (molecular formula: C_4_H_8_O_2_, molecular weight: 88.10, standard quality), Tetrahydrofuran (THF, molecular formula: C_4_H_8_O, molecular weight: 72.11, standard quality), and N, N-Dimethylformamide (DMF, molecular formula: C_3_H_7_NO, molecular weight: 73.10, standard quality) were purchased from Macklin Biochemical Co., Ltd. (Shanghai, China). PNS was purchased from Yunnan Baiyao Group Wenshan Qihua Co., Ltd. (Kunming, China). OTF was purchased from Beijing Qihuang Pharmaceutical Co., Ltd. (Beijing, China). The purities of purchased PNS and OTF were detected by high performance liquid chromatography (HPLC) analysis, and the results showed the purities of purchased OTF and PNS reached the standards (Appendix A). 

The one-layer scaffold, named POP (PLLA+OTF+PNS), was created through thermally induced phase separation (TIPS). At first, 10% PLLA was dissolved in 1,4-dioxane, and water was then added into this solution, stirred at 60 °C for 1 h (1,4-dioxane used in former step:water = 87:13). Then, OTF (8 mg) and PNS (5 mg) were added into the solution and stirred at 60 °C for 20 min. When OTF:PNS = 8:5, the release concentration of both OTF and PNS reached 100 μg/mL so that the proliferation and differentiation of osteoblasts and the proliferation of vascular endothelial cells were promoted in the concentration of 100 μg/mL [22]. The solution was poured into a plastic mold (4 mm diameter) and placed at 30 °C for 1 h, and –20 °C for 1 h, after which the solution was freeze-dried for 24 h. 

The bi-layer scaffold, named PO-PP, included two layers (inner layer: PLLA+ OTF; outer layer: PLLA+ PNS). The inner layer (PO) was prepared by the same method, and the solution was poured into a plastic mold (3 mm diameter). OTF was added into this layer. The outer layer (PP) was created by an electrospun membrane. The 10% PLLA was dissolved in THF and DMF at 60 °C for 1 h. PNS (5 mg) was added into the solution. The resulting solution was electrospun into fibers (voltage: +13, −2 kV; extrusion speed: 0.0025 mm/s). The inner layer was rolled to receive the electrospun membrane by a receiving device (running at 10 rpm) at receiving distance 20 cm. All scaffolds were sterilized by ^60^Co irradiation and stored at room temperature before use.

### 2.2. Characterization of PO-PP and POP Scaffolds

The microstructures of PO-PP and POP scaffolds were observed by field emission-scanning electron microscopy (FE-SEM, FEI Quanta 250 FEG, Hillsboro, OR, USA). The round surfaces of scaffolds were placed on the platform and the pressing speed was 1 mm/min, pressing to 80% length of scaffolds. The maximum compressive strength of the PO layer (3 mm diameter, 16 mm height) and POP scaffold (4 mm diameter, 16 mm height) was assessed through a universal testing machine (EZ-LX, SHIMADZU, Kyoto, Japan). The pycnometer method was used to assess the porosity of the porous structure (PO layer and POP scaffold). The mass *m*_1_ (g) of the scaffolds was weighed. The pycnometer filled with deionized water also likewise weighed *m*_2_ (g); the sample was placed into the pycnometer and soaked adequately. Then, the pycnometer was weighed *m*_3_ (g) after filling up with water. The mass of the remaining water and the pycnometer weighed *m*_4_ (g), which were assessed after the soaked sample taken out. The porosity of the scaffolds (ε) was calculated based on foregoing data, as shown in the equation *ε* = (*m*_3_ – *m*_4_ – *m*_1_)/(*m*_2_ – *m*_4_) × 100%. 

The loaded OTF and PNS were evaluated by a Fourier transform infrared spectroscope (FTIR, FTIR-7600, Lambda Scientific, Rüsselsheim, Germany). The sheared scaffold and potassium bromide were mixed at a ratio of 1:100, pressed, and scanned by FTIR at a scanning range of 4000–400 cm^−1^. 

The release medium was PBS (37 °C) and shaken at 100 rpm. The cumulative release rate of OTF was measured at 283 nm in quartz plate by ultraviolet spectrophotometry equipped with a microplate reader (Thermo Multiskan FC, Waltham, MA, USA). The cumulative release rate of PNS was measured through the vanillin-perchloric acid chromogenic method. After the solvent evaporated, prepared 5% vanillin-glacial acetic acid (0.2 mL) and perchloric acid (0.8 mL) were added. The mixture was heated at 60 °C for 20 min in a thermostatic water bath, cooled for 5 min in a cold water bath, and then 5 mL glacial acetic acid was added. Finally, PNS was measured at 545 nm by a microplate reader.

### 2.3. Animal Grouping and Surgery

Thirty rabbits were used in this experiment. All rabbits were administered 10 μg/kg of lipopolysaccharide (LPS, Sigma-Aldrich, St. Louis, MO, USA) fluid intravenously. Each rabbit was then given an intramuscular injection of 20 mg/kg methylprednisolone (MPS, Macklin, Shanghai, China) every 24 h, which was repeated three times. And then, two weeks later, a femoral head necrosis model was successfully induced, which was reported in our previously published article [23]. The thirty rabbits were randomly divided into three groups 12 weeks after modeling: the PO-PP group, imbedded with PO-PP scaffold; the POP group, imbedded with POP scaffold; and the control group, imbedded with no scaffold. 

The surgery was conducted two weeks after the third MPS injection. Femoral head perforation was only performed on the right side. A 25 mm skin incision was made to expose the right hip joint. A bone tunnel (4 mm diameter, 16 mm depth) from the hole to the femoral head through the mid-axis of femoral neck was created by guiding a C-arm fluoroscope. The PO-PP and POP scaffolds (PO-PP group, 4 mm × 16 mm PO-PP scaffold; POP group, 4 mm × 16 mm POP scaffold; control group, no scaffold) were inserted into the bone tunnels. These rabbits were intramuscularly injected with 20 wu of penicillin twice a day for two weeks. After 12 weeks post-surgery, the experimental rabbits were sacrificed by intravenously injecting 150 mg/kg bodyweight sodium pentobarbital. The whole right femur was resected, and the soft tissues were removed. Finally, the whole right femurs were fixed in 10% neutral formalin for micro-CT scanning, as well as histological, immunohistochemistry, and immunofluorescence analyses. 

### 2.4. Micro-CT Analysis

The harvested femoral head samples were assessed by a micro-CT scanner (Skyscan1076, Bruker, Karlsruhe, Germany) at 40 kVP, 250 μA for 800 ms. The region of interest (ROI, 4 mm diameter in the center of the bone tunnel) was selected and reconstructed by a CT-analyzer. The bone mineral density (BMD, g/cc), bone surface density (BS/TV, 1/mm), percent bone volume (BV/TV, %), trabecular number (Tb.N, 1/mm), and bone volume (BV, mm^3^) were calculated based on the data from the micro-CT scans. 

### 2.5. Histological Analysis

The femoral heads were decalcified with 14% EDTA solution (pH7.2) for 28 days. The decalcified samples were embedded in paraffin and cut into 4 μm sections. The Hematoxylin and Eosin (H&E) (Servicebio, Wuhan, China), Masson trichrome (MT) (Servicebio), Goldner’s trichrome (Servicebio), and Tartrate-resistant acid phosphatase (TRAP) (Servicebio) stainings were executed on the sections. The histological sections were observed with a light microscope (Olympus, Tokyo, Japan).

### 2.6. Immunohistochemistry (IHC) Analysis

The quantity and distribution of the vascular endothelial growth factor (VEGF), osteopontin (OPN), and osteocalcin (OCN) in the bone samples were detected through IHC analysis. The sections were incubated for heat-induced epitope retrieval in citrate buffer at 60 °C, then blocked with 1% hydrogen peroxide/methanol (Sigma-Aldrich, St. Louis, MO, USA) for 30 min at 25 °C. After blocking, the sections were incubated with primary antibodies VEGF (1:100; Novus NBP2-45235, Bio-Techne Corporation, Minneapolis, MN, USA), OPN (1:100; Novus NB110-89062, Bio-Techne Corporation, Minneapolis, MN, USA) and OCN (1:100; Novus NBP2-89037, Bio-Techne Corporation, Minneapolis, MN, USA) overnight at 4 °C. Next, the samples were incubated with goat anti-mouse secondary antibodies (1:200; Servicebio G23301, Wuhan servicebio technology CO., LTD, Wuhan, China) for 1 h at room temperature. The positive staining sites were colored by 3,3’-diaminobenzidine (Dako Liquid DAB, Carpinteria, CA, USA) color solution. Finally, the sections were counterstained with hematoxylin. The sections were then visualized with a light microscope, and the positive staining sites were assessed by Image J software (Image J, V1.8.0, Bethesda, MD, USA). 

### 2.7. Immunofluorescence Assay (IMA)

IMA was performed to detect the quantity and distribution of the platelet endothelial cell adhesion molecule-1 (PECAM-1/CD31) and alkaline phosphatase (ALP). After heat-induced epitope retrieval and blockage, the slides were incubated with primary antibodies for CD31 (1:20; Novus NB600-562, Bio-Techne Corporation, Minneapolis, MN, USA) and ALP (1:100; Novus NB600-540, Bio-Techne Corporation, Minneapolis, MN, USA) overnight at 4 °C. Subsequently, the samples were incubated with goat anti-rabbit secondary antibodies (CD31:1:400, Servicebio GB25303, Wuhan servicebio technology CO., LTD, Wuhan, China; ALP:1:300, Servicebio GB21303, Wuhan servicebio technology CO., LTD, Wuhan, China) for 1 h at room temperature. Ultimately, the samples were counterstained with DAPI (Servicebio) for 10 min at room temperature. Images were captured with a Leitz/Leica TCSSP2 microscope (Leica Lasertechnik GmbH, Heidelberg, Germany) and assessed by ImageJ software (Image J, V1.8.0, Bethesda, MD, USA).

### 2.8. Statistical Analysis

All data were presented as mean ± standard deviation (SD). Differences among the groups were assessed using one-way analysis of variance (ANOVA) with LSD for multiple comparisons in this study. A *p* value < 0.05 was considered as statistically significant. All data analyses were performed using the SPSS 23.0 software (IBM, Armonk, NY, USA).

## 3. Results

### 3.1. Characterization of Scaffolds

The PO-PP scaffold is a bi-layer scaffold including PLLA embedded with OTF (PO, inner layer) and PLLA embedded with PNS (PP, outer layer), which achieves sequential delivery of PNS and OTF; PNS of the outer layer is initially released, followed by the release of OTF from the inner layer. The initially released PNS could promote new vessel formation, which would further enhance new bone formation by providing sufficient nutrition to the bone tissue. The inner OTF was released later, which could promote new bone formation. In this manner, better and faster new bone formation is obtained. Enhancing the osteogenic effect of vascularization (PNS in the outer layer) is the key technique employed by our scaffold. The PO layer has a porous structure, and the PP layer has an electrospinning net. The POP scaffold has a one-layer scaffold, including PLLA embedded with OTF and PNS (POP), which did not obtain sequential release and released OTF and PNS simultaneously. It only had a porous structure (Figure 1A,B). For the PO-PP scaffold, the pore diameter was about 10–38 μm (PO layer), and the electrospinning diameter was about 0.7–1.7 μm (PP layer). For the POP scaffold, the pore diameter ranged within 14–31 μm. Table 1 indicates that porous materials (PO layer, POP scaffold) meet the minimum strength requirement of cancellous bone repair (1 MPa) and the minimum porosity requirement of vessel formation (50%) [24,25]. OTF was successfully loaded into the PO layer of the PO-PP scaffold with hydroxyl groups (3470 cm^−1^) and benzene ring (1654 cm^−1^), which was the characteristic absorption peak of OTF in the FTIR image (Figure 2A). PNS with hydroxyl structures (3470 cm^−1^) and a carbon–carbon double bond (1639 cm^−1^) was successfully loaded into PP layer of PO-PP scaffold in the FTIR image (Figure 2A). In this manner, OTF and PNS were both successfully loaded into the POP scaffold. For cumulative drug release, no difference was found between the PO-PP and POP scaffolds in OTF (Figure 2B). The PNS cumulative release was faster in the PO-PP scaffold than in the POP scaffold, which not only achieves better vessel formation, but also enhances the osteogenic effect of vascularization of the PO-PP scaffold to some degree (Figure 2B). The weight-averaged molecular weight (Mw) of PLLA was about 2 × 10^5^ and slowly degraded in vivo, which was reported in our previous published article [26].

### 3.2. Micro-CT Analysis for Bone Formation Treated by Scaffold Implantation

The micro-CT analysis was conducted twelve weeks after surgery. As shown in Figure 3, the scaffolds were successfully implanted into the femoral head of animals. The reconstructed 3D image indicates that the new bone formation in the PO-PP and POP groups was significantly better than that in the control group; and further PO-PP scaffold exhibits more new bone formation than the POP scaffold. The BMD, BV, and BV/TV of the PO-PP and POP groups were significantly higher than those of the control group (*p* < 0.05); furthermore, the data of BMD, BV, and BV/TV in the PO-PP group show significant differences compared with those in the POP group (*p* < 0.01). The data of Tb.N and BS/TV in the PO-PP and POP groups show significant differences compared with those in the control group (*p* < 0.05). These results indicate that PO-PP and POP scaffolds all have a good osteogenesis effect. The osteogenic capacity of the PO-PP scaffold is superior to that of the POP scaffold, which indicates the advantage of sequential delivery (the key technique of PO-PP scaffold).

### 3.3. Histological Analysis of Bone Repair

In the process of repairing GONFH, vessels delivered nutrients to the femoral head and improved the new bone formation in the necrotic area. In this experiment, newly formed vessel and bone around the tunnel were assessed by HE staining. As shown in Figure 4A, there were more new vessels and trabecular bones filled at the injury site of the femoral head of the PO-PP and POP groups as compared with that of the control group. The repair effect of the PO-PP group was superior to that of the POP group. MT staining was commonly used to identify collagen (blue) and osteoid tissue (red). A mount of collagen fibers and osteoid tissue were filled around the defect regions in the treatment groups (Figure 4B). The therapeutic effect of the PO-PP group was superior to that of the POP group. The results of HE and Masson staining indicated better angiogenic and osteogenic effects of the PO-PP scaffold. It is the contribution of the PO-PP scaffold’s sequential release that enhances the osteogenic effect of vascularization. 

Bone, a dynamic tissue, remodels continuously throughout life, which constitutes an important process for bone repair [27]. Bone remodeling is tightly regulated by the balance between osteoblasts and osteoclasts [27]. Osteoid, a collagen-rich matrix, is produced by osteoblasts and mineralized extracellularly by nanosized calcium phosphate (CaP) [28]. The formed osteoid in the mineralized bone represents the osteogenic capacity. Goldner staining was conducted to distinguish between osteoid tissue (red) and mineralized bone tissue (green). TRAP staining, used to reveal the osteoclast activity, was also performed to assess the osteoclastic bone resorption [29]. The results of Goldner staining (Figure 5A) show a large amount of osteoid tissue around the tunnel, appearing near the implantation site of the treatment groups. The amount of osteoid tissue in the PO-PP group was more than those in the POP group, which indicates better promoting effect of osteogenic activity in the PO-PP scaffold. As shown in Figure 5B, a large quantity of TRAP-positive red cells was detected near the tunnel of the control group, while the defect regions in the PO-PP and POP groups were surrounded with a small quantity of TRAP-positive red cells after implantation. The quantity of TRAP-positive red cells was lower in the PO-PP group than in the POP group, which indicates a better inhibiting effect of osteoclastic activity in the PO-PP scaffold. In summary, the PO-PP scaffold has better effects in promoting osteogenic activity and inhibiting osteoclastic activity, owing to the sequential release of the PO-PP scaffold. Initially released PNS not only promotes vessel formation, but also enhances new bone formation by providing sufficient nutrition to bone tissue.

### 3.4. Distribution and Quantity of VEGF and CD31 in Implantation Sites

Vascularization during bone repair supports the recruitment of circulating osteoblast-lineage cells for osteogenesis [30]. This is the key to the sequential release of the PO-PP scaffold. Therefore, to investigate the effector mechanism of vascularization, VEGF and CD31stainings were performed (Figure 6A and Figure 7A). Positive cells were stained brown for VEGF (IHC), and green for CD31(IMA). The distribution and quantity of VEGF proteins was significantly increased in the treatment groups compared with those in the control group (*p* < 0.01), and furthermore, the PO-PP scaffold showed much higher VEGF protein expression when compared with the POP scaffold (*p* < 0.01) (Figure 6B). The same observation was made on the CD31 protein in the treatment groups, compared with that of the control group around the tunnel after implantation (*p* < 0.01). The PO-PP scaffold also showed significantly higher CD31 protein expression compared with the POP scaffold (*p* < 0.01) (Figure 7B). The higher protein expressions of VEGF and CD31 in the PO-PP scaffold might be attributed to the sequential release of the PO-PP scaffold, where PNS is released first and at a faster release rate. 

### 3.5. Distribution and Quantity of OCN, OPN, and ALP in Implantation Sites

To explore the osteogenesis mechanism, the distribution and quantity of OCN, OPN, and ALP around implantation sites were assessed (Figure 8, Figure 9 and Figure 10). The positive cells were stained brown for OCN and OPN (IHC), and red for ALP (IMA). The distribution and quantity of OCN, OPN, and ALP in the treatment groups were increased significantly compared to those in the control group (*p* < 0.01). Furthermore, the data of OCN, OPN, and ALP expression in the PO-PP scaffold showed significant differences when compared with those in the POP scaffold (*p* < 0.05). This represented the advantage of the sequential release of the PO-PP scaffold, which not only provided the osteogenic effect of vascularization (PP layer), but also the osteogenic effect of OTF (PO layer).

## 4. Discussion

GONFH, characterized by progressive deterioration of the hip joint [31], is a serious complication occurring after long-term or excess administration of clinical glucocorticoids [32]. With the development of GONFH, the femoral head will collapse, ultimately requiring total hip arthroplasty (THA) [33,34]. About 30–60,000 new cases of GONFH occur in the United States every year [35], and approximately 8.12 million people suffer from GONFH in China [36]. GONFH can create substantial financial, physical, and emotional burdens for patients and their families. 

Glucocorticoids have an inhibitory function on angiogenesis in the femoral head, such that synthetic glucocorticoid treatment causes vascularity disruption at the site [37]. Osteogenesis requires an adequate blood supply [38], and vascular damage easily results in an inability to deliver nutrients to the femoral head [1]. Angiogenesis and osteogenesis are closely related to femoral head repair [4]. To achieve the osteogenic effect of vascularization, we aim to design a sequentially drug-releasing scaffold. PLLA, a biodegradable and biocompatible scaffold material, has been widely used as carrier for delivering drugs or bioactive factors in orthopedic treatment [39]. PNS, the main ingredient of Panax Notoginseng, is able to promote angiogenesis. It thus prevents the loss of vascular endothelial cells [40]. OTF, the major component of Drynariae Rhizoma, promotes osteogenesis. Several studies showed that OTF promotes bone formation in large tibial defects (LTDs) [18], and thus increases the bone density and mechanical strength of tail suspension rats [41]. In this study, we designed a sequentially released bi-layer PLLA scaffold, where the inner layer contained OTF, and the outer layer contained PNS. PNS in the outer layer was initially released to promote vessel formation and, more importantly, to enhance the osteogenic effect of vascularization (the key technique employed in our study). The OTF in the inner layer was released later to promote new bone formation. We also made a one-layer PLLA scaffold containing OTF and PNS, which were successfully loaded into scaffolds. The PO-PP and POP scaffolds released PNS and OTF steadily and durably. The PNS release was faster in the PO-PP scaffold than in the POP scaffold, which also supported the osteogenic effect of vascularization.

After scaffold implantation, we assessed new bone formation through a micro-CT. The results of reconstructed 3D image showed that the new bone formation in the treatment groups was significantly better than that in the control group; and the PO-PP scaffold had more new bone formation than the POP scaffold. The data of BMD, BV, and BV/TV in the treatment groups were significantly higher than those in the control group with *p* < 0.05; and the data of BMD, BV, and BV/TV in the PO-PP group were significantly higher than those in the POP group with *p* < 0.01. These results indicated that both scaffolds had good osteogenesis effect, while the osteogenic effect of the PO-PP scaffold was superior to that of the POP scaffold. We observed the formation of new vessels and new bone via HE staining. The results showed that both scaffolds exhibited more vessel and bone formation around the defect when compared with the control group, and more vessel and bone formation was observed in the PO-PP scaffold than in the POP scaffold. Because bone formation is tightly regulated by the balance between osteoblasts and osteoclasts [27], we observed the osteogenic activity by Masson staining, Goldner staining, and the osteoclastic activity by TRAP staining. The results of these stainings indicate that both scaffolds promote osteogenic activity and inhibit osteoclastic activity. Interestingly, superior effects (promoting osteogenic activity and inhibiting osteoclastic activity) were found in the PO-PP scaffold compared with the POP scaffold.

In the process of promoting bone repair through angiogenesis and osteogenesis, the PO-PP scaffold not only promotes more vessel formation via the faster-released PNS, but the PNS in the outer layer enhanced the osteogenic effect of vascularization based on the sequential release system. Therefore, sequential release, which represent the key technology in our study, not only achieved more angiogenesis, but also indirectly promoted more new bone formation. Moreover, the OTF in the inner layer also promoted new bone formation. The one-layer scaffold (POP) obtained a slow PNS release. Furthermore, the simultaneous release of PNS and OTF lost the advantage of the sequential release, thus depriving PNS of its role in indirect osteogenesis. Therefore, a sequential release is the key technology in the PO-PP scaffold to achieve better bone repair. 

To explore the possible mechanisms of angiogenesis and osteogenesis, the distribution and quantity of key proteins were evaluated through IHC and IMA. VEGF (a homodimeric disulfide bound glycoprotein) promotes endothelial growth [42]. CD31 (the most abundant constitutive co-signaling receptor glycoprotein on endothelial cells) is highly expressed at endothelial cell–cell junctions [43], and forms the first-line of cellular contact with the blood [44]. The results in our study showed that the protein expressions of VEGF and CD31 in the treatment groups were significantly higher than those in the control group with *p* < 0.01, and the protein expressions of VEGF and CD31 in the PO-PP scaffold were significantly higher than those in the POP scaffold with *p* < 0.01. The enhancing effect of the PO-PP scaffold might be attributed to the sequential release of the PO-PP scaffold where PNS is released first and at a faster release rate. OCN synthesized by osteoblasts is known for its strong binding affinity to bone minerals [45]. OPN, which binds to extracellular calcium via acidic (aspartate and glutamate) and phosphorylated (serine and threonine) residues, plays a key role in nucleating bone minerals [46]. ALP (a homodimer anchored to the membrane of osteoblasts) is essential for biomineralization [47]. The results indicated that the protein expressions of OCN, OPN, and ALP in the treatment groups were significantly higher than those in the control group with *p* < 0.01, and the protein expressions of OCN, OPN, and ALP in the PO-PP scaffold were significantly higher than those in the POP scaffold with *p* < 0.05. The promotion function of the PO-PP scaffold was better than that of the POP scaffold. Based on the sequential release advantage of the bi-layer scaffold, the PO-PP scaffold had a better angiogenic and osteogenic effect. Research into the mechanisms further found that the PO-PP scaffold achieved the above effects by promoting the expression of angiogenic and osteogenic proteins. 

## 5. Conclusions

We fabricated a sequentially released bi-layer scaffold called a PO-PP scaffold. In this PO-PP scaffold, PNS was initially released, followed by the release of OTF to obtain the osteogenic effect of angiogenesis (the key advantage of our scaffold). Compared with the POP scaffold (with concomitant release of OTF and PNS), the PO-PP scaffold had better effects in promoting new bone formation and blood vessel formation, promoting osteoid tissue formation, and inhibiting osteoclastic activity. Upregulating the expression of angiogenic proteins (VEGF and CD31) and osteogenic proteins (OCN, OPN and ALP) might be the possible mechanism. Therefore, the sequentially released PO-PP scaffold proposes a new therapy for bone repair. Based on the osteogenic effect of angiogenesis of the sequentially released PO-PP scaffold, we want to apply this theory to osteoporotic bone defect in rats (different disease and different species) in future to explore the possible effects and mechanisms. 

## Figures and Tables

**Figure 1 jfb-14-00031-f001:**
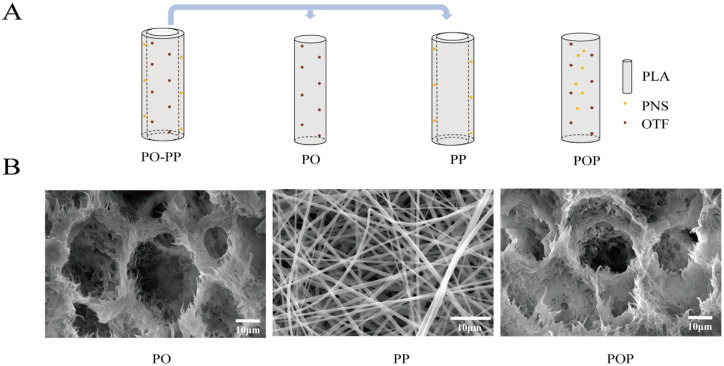
Scaffold characteristics. (**A**) Structural diagrams of PO-PP and POP scaffolds. PO-PP scaffold is a bi-layer scaffold, including inner layer (PO layer: PLLA embedded with OTF) and outer layer (PP layer: PLLA embedded with PNS). POP scaffold is a one-layer scaffold (PLLA embedded with OTF and PNS). (**B**) SEM images of PO-PP and POP scaffolds. PO layer exhibits porous structure at magnifications of 4500×; PP layer shows electrospinning net at magnifications of 2500×; POP scaffold has a porous structure at magnifications of 4500×.

**Figure 2 jfb-14-00031-f002:**
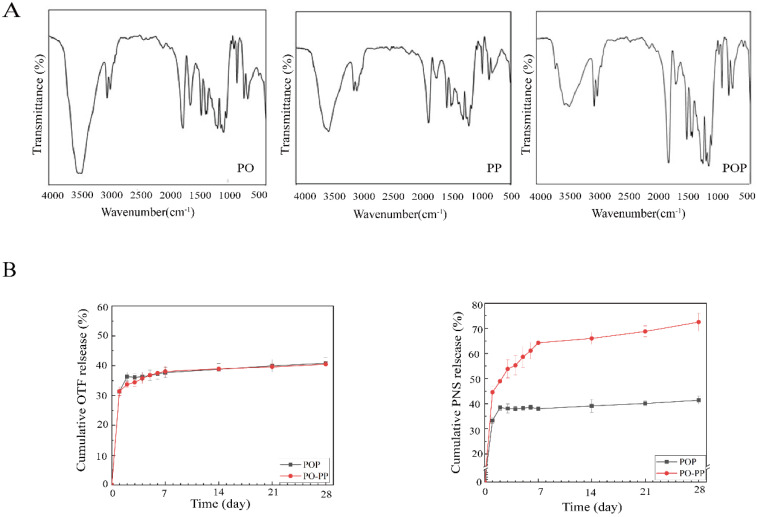
Scaffold characteristics. (**A**) FTIR diagram of scaffolds including PO layer, PP layer, and POP scaffold. (**B**) Cumulative release profiles of OTF and PNS from POP and PO-PP scaffolds at different times; data are presented as mean ± SD, n = 10.

**Figure 3 jfb-14-00031-f003:**
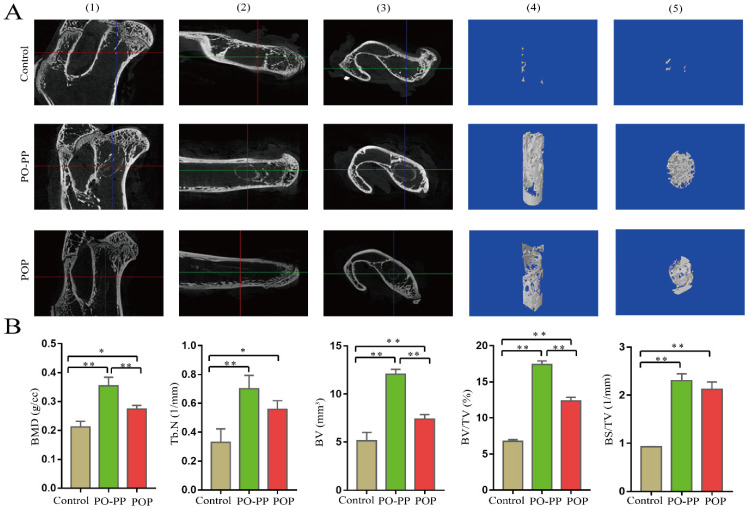
New bone formation in bone tunnel. (**A**) Scanning image (black): the focus of the red and green lines depicts the scaffold, (1) Coronal plane, (2) Sagittal plane, (3) Cross-section. Representative 3D micro-CT images (blue) within a ROI of central 3 mm in diameter of the bone tunnel, (4) Coronal plane, (5) Cross-section. (**B**) Quantitative analysis of new bone in bone tunnel: BMD, Tb.N, BV, BV/TV, BS/TV. n = 10; Error bar represents SD. ANOVA test with LSD analysis is used for statistical analysis; * *p* < 0.05 vs. control group or POP group; ** *p* < 0.01 vs. control group or POP group.

**Figure 4 jfb-14-00031-f004:**
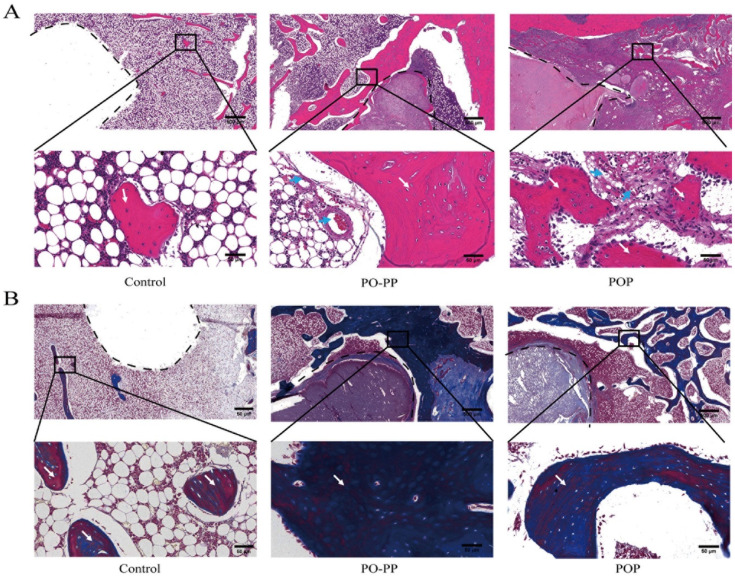
Histological analysis of bone repair after scaffold implantation. (**A**) H&E staining of decalcified sections (bone tunnel, black circle) at magnifications of 20× and 200×. The 200× images represent higher magnifications of areas within black boxes. White arrow indicates trabecular bone, blue arrow indicates vessel, bars = 500 and 50 μm for 20× and 200×, respectively. (**B**) Masson trichrome staining of decalcified sections (bone tunnel: black circle) at magnifications of 20× and 200×. 200× images are higher magnifications of the areas within black boxes. White arrow indicates collagen (blue) and osteoid tissue (red), bars = 500 and 50 μm for 20× and 200×, respectively.

**Figure 5 jfb-14-00031-f005:**
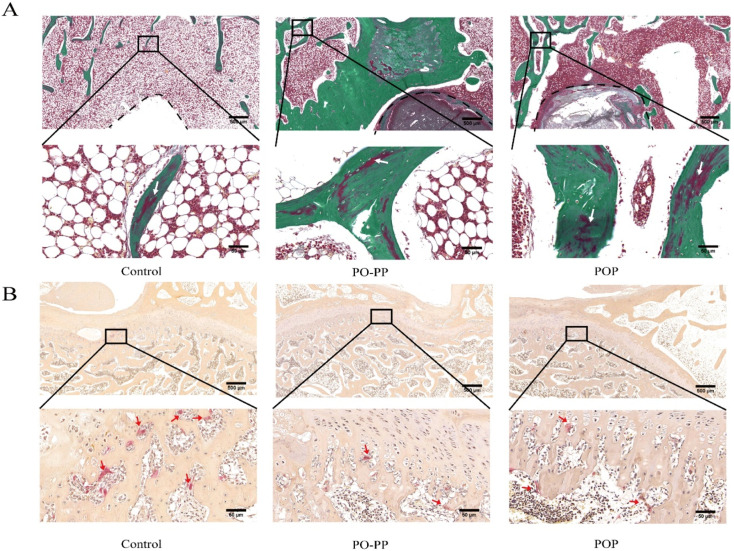
Histological analysis of bone repair after scaffold implantation. (**A**) Goldner staining of decalcified sections (bone tunnel: black circle) at magnifications of 20× and 200×; 200× images are higher magnifications of areas within black boxes. White arrow indicates osteoid tissue, bars = 500 and 50 μm for 20× and 200×, respectively. (**B**) TRAP staining of decalcified sections at magnifications of 20× and 200×, 200× images are higher magnifications of areas within black boxes. Red arrow indicates osteoclast, bars = 500 and 50 μm for 20× and 200×, respectively.

**Figure 6 jfb-14-00031-f006:**
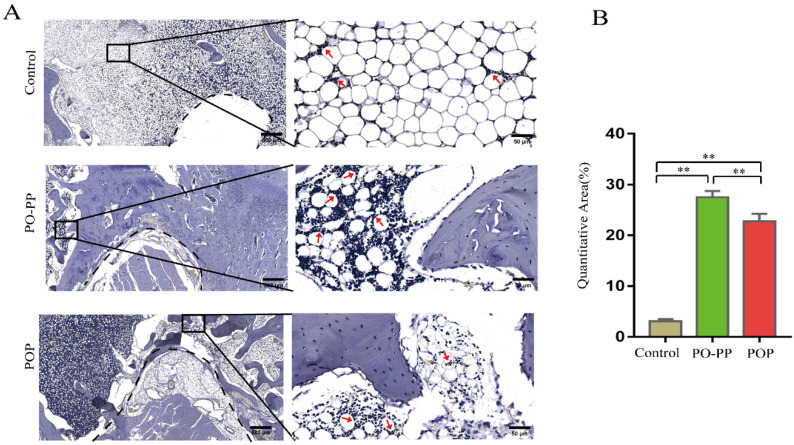
Immunohistochemistry staining of VEGF after scaffold implantation. (**A**) Distribution and quantity of VEGF in decalcified sections (bone tunnel, black circle) at magnifications of 20× and 200×. 200× images are higher magnifications of areas within black boxes. Red arrow indicates VEGF positive cells, bars = 500 and 50 μm for 20× and 200×, respectively. (**B**) Quantitative analysis of VEGF expression. n = 10 in each group, error bar represents SD. ANOVA test with LSD analysis was used for statistical analysis, * *p* < 0.05 vs. control group or POP group; ** *p* < 0.01 vs. control group or POP group.

**Figure 7 jfb-14-00031-f007:**
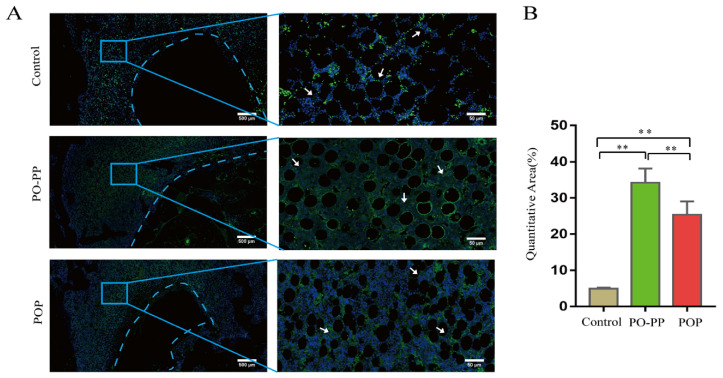
Immunofluorescence staining of CD31 after scaffold implantation. (**A**) Distribution and quantity of CD31 in decalcified sections (bone tunnel: blue circle) at magnifications of 20× and 200×. 200× images are higher magnifications of areas within blue boxes. White arrow indicates CD31 positive cells, bars = 500 and 50 μm for 20× and 200×, respectively. (**B**) Quantitative analysis of CD31 expression. n = 10 in each group, error bar represents SD. ANOVA test with LSD analysis was used for statistical analysis; * *p* < 0.05 vs. control group or POP group; ** *p* < 0.01 vs. control group or POP group.

**Figure 8 jfb-14-00031-f008:**
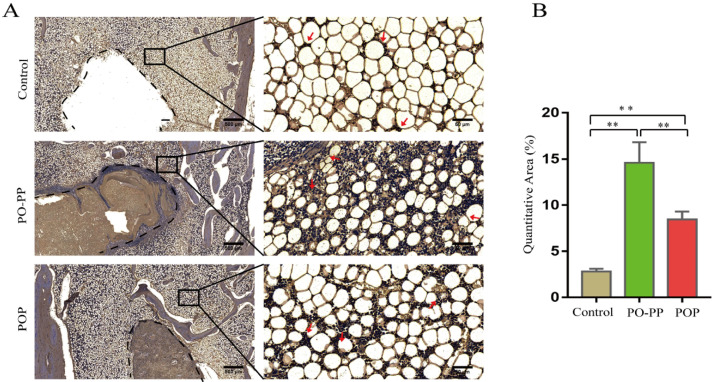
Immunohistochemistry staining of OCN after scaffold implantation. (**A**) Distribution and quantity of OCN in decalcified sections (bone tunnel: black circle) at magnifications of 20× and 200×. 200× images are higher magnifications of areas within black boxes. Red arrow indicates OCN positive cells, bars = 500 and 50 μm for 20× and 200×, respectively. (**B**) Quantitative analysis of OCN expression. n = 10 in each group, error bar represents SD. ANOVA test with LSD analysis was used for statistical analysis, * *p* < 0.05 vs. control group or POP group; ** *p* < 0.01 vs. control group or POP group.

**Figure 9 jfb-14-00031-f009:**
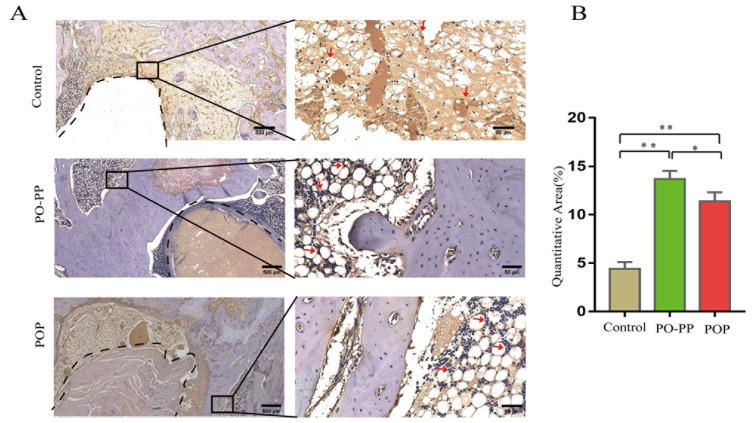
Immunohistochemistry staining of OPN after scaffold implantation. (**A**) Distribution and quantity of OPN in decalcified sections (bone tunnel, black circle) at magnifications of 20× and 200×. 200× images are higher magnifications of areas within black boxes. Red arrow indicates OPN positive cells, bars = 500 and 50 μm for 20× and 200×, respectively. (**B**) Quantitative analysis of OPN expression. n= 10 in each group, error bar represents SD. ANOVA test with LSD analysis was used for statistical analysis, * *p* < 0.05 vs. control group or POP group, ** *p* < 0.01 vs. control group or POP group.

**Figure 10 jfb-14-00031-f010:**
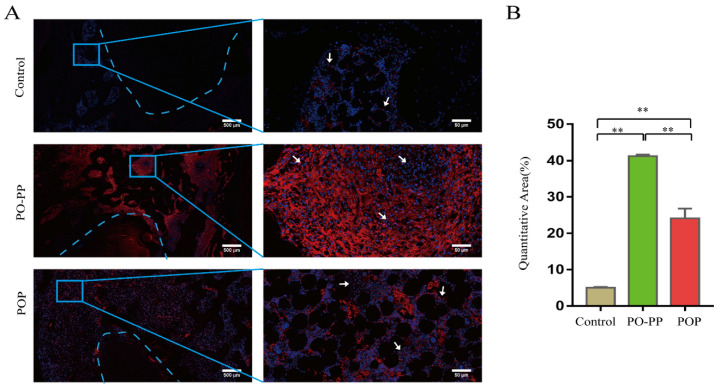
Immunofluorescence staining of ALP after scaffold implantation. (**A**) Distribution and quantity of ALP expression in decalcified sections (bone tunnel: blue circle) at magnifications of 20× and 200×. 200× images are higher magnifications of areas within blue boxes. White arrow indicates ALP positive cells, bars = 500 and 50 μm for 20× and 200×, respectively. (**B**) Quantitative analysis of ALP expression. n = 10 in each group, error bar represents SD. ANOVA test with LSD analysis was used for statistical analysis, * *p* < 0.05 vs. control group or POP group; ** *p* < 0.01 vs. control group or POP group.

**Table 1 jfb-14-00031-t001:** Porosity and maximum compressive strength of PO(PO-PP) and POP.

Scaffold	Porosity (%)	Maximum Compressive Strength (MPa)
PO layer in PO-PP scaffold	89.56 ± 2.15	1.46 ± 0.13
POP scaffold	91.06 ± 2.13	1.34 ± 0.12

## Data Availability

The data presented in this study are available on request from the corresponding author.

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
