# Peer review of "Sequential Release of Panax Notoginseng Saponins and Osteopractic Total Flavone from Poly (L-Lactic Acid) Scaffold for Treating Glucocorticoid-Associated Osteonecrosis of Femoral Head"

_jfb, 2023, doi:10.3390/jfb14010031_

Round 1

Reviewer 1 Report

The work entitled "Sequential release of panax notoginseng saponins and osteopractic total flavone from Poly (L-lactic acid) scaffold for treating glucocorticoid-associated osteonecrosis of femoral head" correlates with the objectives and scope of the journal. The authors performed a thorough characterization of the materials and in vivo tests show convincing results.  According to this reviewer's criteria the work can be published in the present form. 

Author Response

Reviewer 1

Comments and Suggestions for Authors

The work entitled "Sequential release of panax notoginseng saponins and osteopractic total flavone from Poly (L-lactic acid) scaffold for treating glucocorticoid-associated osteonecrosis of femoral head" correlates with the objectives and scope of the journal. The authors performed a thorough characterization of the materials and in vivo tests show convincing results.  According to this reviewer's criteria the work can be published in the present form. 

Response: Thanks very much for reviewing the manuscript. We have carefully checked and revised the grammar and writing form of the article.

Reviewer 2 Report

1.    The abstract of the article should be supported with some quantitative determinations (numerical values found during different experiments).

2.    Maintain uniform style in writing the digit/numerical values and its unit with at least one spacing between them. For example, “5 mg” (correct) not 5mg (wrong practice). Also not “1mm / min” but “1 mm/min”

3.    Correct the symbol of degree centigrade throughout the manuscript and negative logarithm of H+ ion is as known pH; it is written as “pH” not PH.

4.    Recheck the equation for porosity (ε) calculation.

5.    No significant difference was found in the SEM images of PO and POP scaffolds (as mentioned in Figure 1B). Also the porous structure of the developed scaffolds should be check at the same magnifications to see any difference in their morphological structure. Why PP layer was checked at 2500× magnifications while others were checked at 4500-times?

6.    How the authors concluded that the PNS with hydroxyl structures (3470 cm-1) and carbon-carbon double bond (1639 cm-1) was successfully loaded into PP layer of PO-PP scaffold in the FTIR, while as shown in the images (Figure 2A), no significant difference was noted or the peaks of particular functional groups appeared at almost save wavenumber with slight fluctuations in the intensity. If any material is loaded into any scaffolds/structure, in this case the inherent peaks of different functional groups of the loaded/ encapsulated materials should not appear in the FTIR spectra of the scaffold. Also the particular wavenumbers (cm-1) in the FTIR spectra of the tested products are not in these images.

7.    As shown in Figure 2B, the release characteristic of PO-PP scaffold and POP scaffold at different times, what could be the rationale of this experiment, what the authors wanted to convey through this experiment?

8.    The degradation of the different scaffolds must be seen at the particular pH at which the release experiment was performed.  

9.    Recheck the statement in the caption of Figure 3. “*P < 0.05 vs. control group or POP group; **P < 0.01 vs. control group or POP group”, is it correct?

10. Conclusions of the article must be supported with some data found during the experiments.

11. At the end of conclusion there must a few sentences regarding the future direction of the present work.

Author Response

Dear Reviewer,

Thanks very much for reviewing this manuscript. Those comments are all valuable and very helpful for revising and improving our paper, as well as the important guiding significance to our researches. We have studied comments carefully and have made correction which we hope meet with approval. The English language and style are improved, and the editorial certificate is uploaded to the system. Revised portion are marked in blue in the paper. The main corrections in the paper and the responds to the reviewer’s comments are as following.

Comments and Suggestions for Authors

Comment1 The abstract of the article should be supported with some quantitative determinations (numerical values found during different experiments).

Response: Thanks for your kind help. According to your kind advice, we have added numerical values found during Micro-CT analysis, immunohistochemistry analysis and immunofluorescence assay in the abstract section of revised manuscript.

Comment 2   Maintain uniform style in writing the digit/numerical values and its unit with at least one spacing between them. For example, “5 mg” (correct) not 5mg (wrong practice). Also not “1mm / min” but “1 mm/min”

Response: Thanks very much for your kind help. According to your kind help, we have corrected the wrong format in our revised manuscript, and maintained the uniform style in writing the digit/numerical values and its unit with at least one spacing between them.

Comment 3 Correct the symbol of degree centigrade throughout the manuscript and negative logarithm of Hion is as known pH; it is written as “pH” not PH.

Response: Thanks for your kind help. The symbol of degree centigrade was corrected carefully throughout the manuscript. And we also corrected “PH” to “pH” in our revised manuscript.

Comment 4    Recheck the equation for porosity (ε) calculation.

Response: We have carefully checked the equation for porosity (ε) calculation.

Comment 5 No significant difference was found in the SEM images of PO and POP scaffolds (as mentioned in Figure 1B). Also the porous structure of the developed scaffolds should be check at the same magnifications to see any difference in their morphological structure. Why PP layer was checked at 2500× magnifications while others were checked at 4500-times?

Response: As shown in “Material and methods- Preparation of one-layer and bi-layer scaffolds”, the one-layer scaffold, named POP (PLLA+OTF+PNS) was created through thermally induced phase separation (TIPS); the bi-layer scaffold, named PO-PP, included two layers (inner layer: PLLA+ OTF; outer layer: PLLA +PNS), and the inner layer (PO) was prepared by the same method. Thus, no significant difference was found in the SEM images of PO and POP scaffolds (as mentioned in Figure 1B). The developed scaffold (PO-PP scaffold) included inner PO layer and outer PP layer; inner PO layer was porous structure, outer PP layer was electrospinning net. The detailed porous structure of PO layer can be better observed at 4500× magnifications than that was observed at 2500× magnifications. Due to the porous structure, we preferred to choose 4500× magnifications to show the detailed porous structure of PO layer clearly. Due to structure difference between porous structure and electrospinning net, the detailed structure of electrospinning net in PP layer could be observed clearly at 2500× magnifications. Based on electrospinning net, PP layer was at 2500× magnifications. In this way, we could not only see the detailed morphological structures of PO layer and PP layer clearly, but also observe the difference in their morphological structures.

Comment 6 How the authors concluded that the PNS with hydroxyl structures (3470 cm-1) and carbon-carbon double bond (1639 cm-1) was successfully loaded into PP layer of PO-PP scaffold in the FTIR, while as shown in the images (Figure 2A), no significant difference was noted or the peaks of particular functional groups appeared at almost save wavenumber with slight fluctuations in the intensity. If any material is loaded into any scaffolds/structure, in this case the inherent peaks of different functional groups of the loaded/ encapsulated materials should not appear in the FTIR spectra of the scaffold. Also the particular wavenumbers (cm-1) in the FTIR spectra of the tested products are not in these images.

Response: Panax notoginseng saponins (PNS) is the main ingredient of Panax Notoginseng (a medicinal herb). The detailed molecular structure of PNS was shown in Figure 1(below). Compared with PLLA, the FTIR image of PP layer (PNS + PLLA) showed new peaks included 3470 cm-1 and 1639 cm-1. PNS, as shown in Figure 1, included many hydroxy structures (3470 cm-1) and carbon-carbon double bond(1639 cm-1). Thus, we have successfully loaded PNS into PLLA to form PP layer, and PNS’s molecular structure has not been destroyed.

Figure 1 The detailed molecular structure of PNS

Comment 7 As shown in Figure 2B, the release characteristic of PO-PP scaffold and POP scaffold at different times, what could be the rationale of this experiment, what the authors wanted to convey through this experiment? What the authors wanted to convey through this experiment?

Response: Thanks very much for your kind help. The bi-layer PO-PP scaffold included two layers (inner layer: PLLA+OTF; outer layer: PLLA+PNS). As outer PP layer was first contact with water and degradation, the PNS encapsulated in outer PP layer was first released to perform angiogenic effect. The degradation of inner PO layer was followed by the degradation of outer PP layer, and as the degradation of inner PO layer, the OTF was released later to perform osteogenic effect. In one-layer POP scaffold (PLLA+OTF+PNS), OTF and PNS released together. The results in Fig.2B showed the sequential release characteristic from PO-PP scaffold (PNS in outer layer released initially, followed by OTF in inner layer) in contrast to the concomitant release characteristic from POP scaffold. Restoring angiogenesis is not only essential for vessel formation, but also crucial for osteogenesis. Osteogenesis requires an adequate blood supply. Angiogenesis and osteogenesis, closely related to each other, are both crucial for bone repair. In order to achieve the osteogenic effect of vascularization, we fabricated this bi-layer PO-PP scaffold included two layers (inner layer: PLLA+ OTF; outer layer: PLLA +PNS). PNS in outer PP layer was initially released to promote vessel formation, and more importantly to enhance the osteogenic effect of vascularization (the key technique employed in our study). OTF in the inner PO layer was released later for promoting new bone formation. The one-layer POP scaffold has the concomitant release characteristic (PNS and OTF were released together). We used one-layer POP scaffold as comparation to confirm the osteogenic effect of vascularization of the sequentially released bi-layer (PO-PP) scaffold.

Comment 8   The degradation of the different scaffolds must be seen at the particular pH at which the release experiment was performed.  

Response: The bi-layer PO-PP scaffold included two layers (inner layer: PLLA+ OTF; outer layer: PLLA +PNS), and the one-layer POP scaffold just has one layer (PLLA+OTF+PNS). The PO-PP scaffold and POP scaffold were both used PLLA to encapsulate drugs. The pH value was manipulated by the degradation of PLLA. In our previous published article “Biodegradable Magnesium-Incorporated Poly (L-lactic acid) Microspheres for Manipulation of Drug Release and Alleviation of Inflammatory Response”, we have found that the media pH (7.4) dropped to a value close to 6.8 after 28 days of PLLA degradation in vitro experiment, due to acidic byproducts generated in the process of PLLA hydrolytic degradation. Thus, the pH value of our study ranged from 7.4 to 6.8, and the drug release experiments of PO-PP scaffold and POP scaffold were performed on it.

Comment 9   Recheck the statement in the caption of Figure 3. “*< 0.05 vs. control group or POP group; **< 0.01 vs. control group or POP group”, is it correct?

Response: Thanks very much for your kind help. We have carefully rechecked the statement in the caption of Figure 3. “*< 0.05 vs. control group or POP group; **< 0.01 vs. control group or POP group”, and the statement is corrected.

Comment 10 Conclusions of the article must be supported with some data found during the experiments.

Response: According to your kind suggestion, we have revised the conclusion section in our revised manuscript.

Comment 11 At the end of conclusion there must a few sentences regarding the future direction of the present work.

Response: According to your kind suggestion, we have added the content regarding the future direction of the present work in conclusion section in our revised manuscript.

Reviewer 3 Report

Thank you very much for opportunity work on this manuscript. In my opinion will be vere interested to readers. I suggest to include discussion regarding errors and data significance. Thank you 

Author Response

Dear Reviewer,

Thanks very much for reviewing this manuscript. Those comments are all valuable and very helpful for revising and improving our paper, as well as the important guiding significance to our researches. We have studied comments carefully and have made correction which we hope meet with approval. The English language and style are improved, and the editorial certificate is uploaded to the system. Revised portion are marked in blue in the paper. The main corrections in the paper and the responds to the reviewer’s comments are as following.

Comments and Suggestions for Authors

Comment Thank you very much for opportunity work on this manuscript. In my opinion will be vere interested to readers. I suggest to include discussion regarding errors and data significance. Thank you 

Response: Thanks very much for your kind advice, we have added data significance in discussion section in our revised manuscript.

Round 2

Reviewer 2 Report

The comments and suggestions have been take care by the authors. 

Author Response

Comments and Suggestions for Authors

The comments and suggestions have been take care by the authors. 

Thanks very much for your kind advice. This manuscript has been carefully revised according to all the suggestions of reviewer.